# Dehydrogenative reagent-free annulation of alkenes with diols for the synthesis of saturated O-heterocycles

Chen-Yan Cai[1] & Hai-Chao Xu [1]

Dehydrogenative annulation reactions are among the most straightforward and efficient approach for the preparation of cyclic structures. However, the applications of this strategy for the synthesis of saturated heterocycles have been rare. In addition, reported dehydrogenative bond-forming reactions commonly employ stoichiometric chemical oxidants, the use of which reduces the sustainability of the synthesis and brings safety and environmental issues. Herein, we report an organocatalyzed electrochemical dehydrogenative annulation reaction of alkenes with 1,2- and 1,3-diols for the synthesis of 1,4-dioxane and 1,4-dioxepane derivatives. The combination of electrochemistry and redox catalysis using an organic catalyst allows the electrosynthesis to proceed under transition metal- and oxidizing reagent-free conditions. In addition, the electrolytic method has a broad substrate scope and is compatible with many common functional groups, providing an efficient and straightforward access to functionalized 1,4-dioxane and 1,4-dioxepane products with diverse substitution patterns.

[1] State Key Laboratory of Physical Chemistry of Solid Surfaces, Innovation Center of Chemistry for Energy Materials, and College of Chemistry and Chemical Engineering, Xiamen University, Xiamen 361005, People's Republic of China. Correspondence and requests for materials should be addressed to H.-C.X. (email: haichao.xu@xmu.edu.cn)

There has been mounting evidence to suggest that the number of saturated carbons and chiral centers in an organic molecule correlate strongly to its clinical prospect[1–4]. Because of this, saturated heterocycles have become increasingly crucial scaffolds for the development of new pharmaceutical compounds. However, unlike heteroaromatics, which can be synthesized conveniently via a variety of cross-coupling reactions, functionalized saturated heterocyclic ring systems have remained challenging to produce[5–7]. For example, the generation of 1,4-dioxane derivatives, which are prevalent in natural products and bioactive compounds (Fig. 1a)[8–13], usually requires a lengthy synthetic procedure and/or complex starting materials that are themselves hard to obtain[8–10,14,15].

Annulation reactions, in which two bonds are formed in a single step, are among the most-efficient methods for the synthesis of cyclic compounds[16]. Particularly, dehydrogenative annulation reactions via X–H (X=C or heteroatom) functionalization provide straightforward access to cyclic scaffolds from easily available substrates[17–19]. Although conventional dehydrogenative annulation reactions often involve the use of stoichiometric chemical oxidants, recent advances in organic electrochemistry[20–30] have led to the development of safer and more environmentally sustainable alternatives that operate under oxidant-free conditions[31–38]. However, to the best of our knowledge, the synthesis of saturated heterocycles with 1,4-diheteroatoms through alkene annulation reactions has not been reported. Although the intramolecular anodic dioxygenation of

heteroatom-substituted alkenes proceeds efficiently[39,40], the intermolecular dimethoxylation of styrenyl alkenes with MeOH under similar conditions resulted in a mixture of compounds generated from 1,2-addition and alkoxydimerization (Fig. 1b)[41,42], probably owing to the relatively high concentration of the alkene radical cation intermediates that were formed on the electrode surface. Moreover, the dimethoxylated product can undergo oxidative decomposition via C–C bond cleavage[43–46].

To minimize the side reactions mentioned above, we argue that the use of redox catalysis[47–58] can facilitate the formation of the desired radical cation in the bulk solution and reduce the electrode potential. Herein, we report a triarylamine-catalyzed electrochemical dehydrogenative annulation reaction of alkenes with 1,2- and 1,3-diols for the synthesis of 1,4-dioxane and 1,4-dioxepane scaffolds (Fig. 1c).

## Results

**Reaction optimization.** The annulation of 1,1-diphenylethene **1** and ethylene glycol **2** was chosen as the model reaction for optimization (Table 1). The electrolysis was conducted in an undivided cell equipped with a reticulated vitreous carbon (RVC) anode and a platinum plate cathode. The highest yield of the 1,4-dioxane product **4** was 91%, achieved when the reaction system consisted of triarylamine $(2,4\text{-Br}_2\text{C}_6\text{H}_3)_3\text{N}$ (**3**)[46,59] $(E_{\text{p}/2} = 1.48$ V vs SCE) as the redox catalyst, $^i\text{PrCO}_2\text{H}$ as acidic additive and an

**a** Selected bioactive 1,4-dioxane-containing compounds

Muscarinic acetylcholine receptor antagonist

R = H, NMDA antagonist
R = Bn, σ₁ -ligand

MK-2461
(c-Met inhibitor, anti cancer)

MMP-13 Inhibitor

**b** Oxidation of styrenyl alkenes using direct electrolysis

1,2-addition

Alkoxydimerization

C-C cleavage

**c** This work: Electrochemical dehydrogenative annulation of alkenes with diols using a redox catalyst

Alkenes    1,2- and 1,3-diols    10 mol % NAr₃ Undivided cell    6- and 7-membered rings    + H₂

**Fig. 1** Reaction design. **a** Selected bioactive molecules containing the 1,4-dioxane moiety. **b** Oxidation of styrenyl alkenes via direct electrolysis. **c** Synthesis of O-heterocycles via annulation reactions of alkenes with diols

**Table 1 Optimization of reaction conditions[a]**

| Entry | Deviation from standard conditions | Yield of 4 (%)[b] |
|---|---|---|
| 1 | None | 91[c] |
| 2 | (4-BrC$_6$H$_4$)$_3$N (**3a**) as the catalyst | 5 (65) |
| 3 | (4-MeO$_2$CC$_6$H$_4$)$_3$N (**3b**) as the catalyst | 44 (38) |
| 4 | **3c** as the catalyst | 38 (25) |
| 5 | No **3** | 58[d] (7) |
| 6 | Reaction at RT | 63 |
| 7 | 2 equiv of **2** | 28[e] |
| 8 | No $^i$PrCO$_2$H | 60 |
| 9 | AcOH as the acid | 77 |
| 10 | EtCO$_2$H as the acid | 87 |
| 11 | CF$_3$CO$_2$H as the acid | 78 |
| 12 | TsOH•H$_2$O as the acid | 80 |
| 13 | Pt plate (1 cm × 1 cm) as anode | 87 |
| 14 | Graphite plate (1 cm × 1 cm) as anode | 66 |

[a]Reaction conditions: reticulated vitreous carbon (RVC) anode (1 cm × 1 cm × 1.2 cm), Pt plate cathode (1 cm × 1 cm), **1** (0.2 mmol), **2** (0.5 mL, 9 mmol), MeCN (5.5 mL), Et$_4$NPF$_6$ (0.2 mmol), 12.5 mA ($j_{anode}$ = 0.16 mA cm$^{-2}$), 1.6 h
[b]Determined by $^1$H NMR analysis using 1,3,5-trimethoxybenzene as the internal standard. Unreacted **1** in parenthesis
[c]Yield of isolated **4**
[d]16% of **5** and 6% of **6**
[e]8% of **5**

excess of **2** in refluxing MeCN (entry 1). Other redox mediators that had a lower oxidation potential than **3**, such as (4-BrC$_6$H$_4$)$_3$N (**3a**, $E_{p/2}$ = 1.06 V vs SCE), (4-MeO$_2$CC$_6$H$_4$)$_3$N (**3b**, $E_{p/2}$ = 1.26 V vs SCE) and the imidazole derivative **3c**[53] ($E_{p/2}$ = 1.19 V vs SCE), were less catalytically effective (entries 2–4). When no catalyst was used, the yield of **4** dropped to 58% and two additional byproducts, **5** and **6**, were obtained in 16 and 6% yields, respectively (entry 5). Running the reaction at RT (entry 6), with only two equiv of **2** (entry 7), in the absence of $^i$PrCO$_2$H (entry 8), or with an alternative acidic additive such as AcOH (entry 9), EtCO$_2$H (entry 10), CF$_3$CO$_2$H (entry 11), and TsOH (entry 12), all led to a significant decrease in reaction efficiency. The replacement of RVC with an anode material that had a smaller surface area, such as Pt (entry 13) or graphite (entry 14), also showed a detrimental effect on the formation of **4**.

**Substrate scope**. We explored the substrate scope under the optimized conditions by first varying the substituents on the alkene (Table 2). The reaction was shown to be broadly compatible with different 1,1-diphenyl alkenes bearing substituents of diverse electronic properties at the para- or meta-position of the benzene ring (**7**–**14**). Meanwhile, alkenes functionalized with a 2-thiophenyl or 2-thiazolyl group were also tolerated (**15**, **16**), albeit with slightly decreased reactivity. When a trisubstituted olefin whose C–C double bond was embedded in a five or six-membered ring was employed, the reaction afforded the cis-fused products (**17**–**19**) as the only diastereomer. However, the employment of a seven-membered cyclic alkene resulted in the generation of a mixture of diastereomers (**20**, dr = 1.8:1). The structure of the minor diastereomer of **20** was subsequently confirmed by X-ray crystallographic analysis. Trisubstituted acyclic alkenes (**21**–**31**)

bearing a halogen (**23**, **24**), free alcohol (**25**), silyl ether (**26**), tosylate (**27**), ester (**28**), sulfonamide (**29**, **30**), or phthalimide (**31**) were all found to be suitable substrates. Furthermore, dioxanes **32** and **33** could be obtained in moderate yields from their corresponding enyne and diene precursors, respectively. It is worth emphasizing that a previous attempt at the anodic reaction of dienes with ethylene glycol generated a mixture of addition and dimerization products instead of the annulation product[60]. Current limitation of the annulation reaction included the inefficient reaction of α-methylstyrene (**34**, 19% yield) and the complete failure of styrene (**35**) and a tetrasubstituted alkene (**36**)

On the other hand, ethylene glycol could be replaced with other vicinal diols such as (2R,3R)-(−)-2,3-butanediol (**37**), cis-1,2-cyclohexanediol (**38**), 2-methyl-1,2-propanediol (**39**, **40**), 2,3-dimethyl-2,3-butanediol (**41**), and 1,3-diols (**42**–**45**). Notably, the unsymmetrically substituted diol 2-methyl-1,2-propanediol reacted regioselectively with different trisubstituted alkenes to afford **39** and **40** bearing two tetrasubstituted carbon centers.

The electrochemical annulation reaction could be conducted on a gram scale as demonstrated by the preparation of 2.5 g of **27** in 80% yield from the annulation of the alkene **46** and **2** (Fig. 2). To allow the application of a higher current, a large anode 50 times the size of that used for the abovementioned model reaction was employed. Gratifyingly, the reaction was completed in < 1 hour, which provided rapid access to **27**.

**Discussion**

A reaction mechanism was proposed in Fig. 3a. The triarylamine catalyst **3** ($E_{p/2}$ = 1.48 V vs SCE) is first anodically oxidized into the radical cation **I**, which in turn oxidizes the alkene substrate **1** ($E_{p/2}$ = 1.68 V vs SCE) through single-electron transfer to furnish

## Table 2 Substrate scope

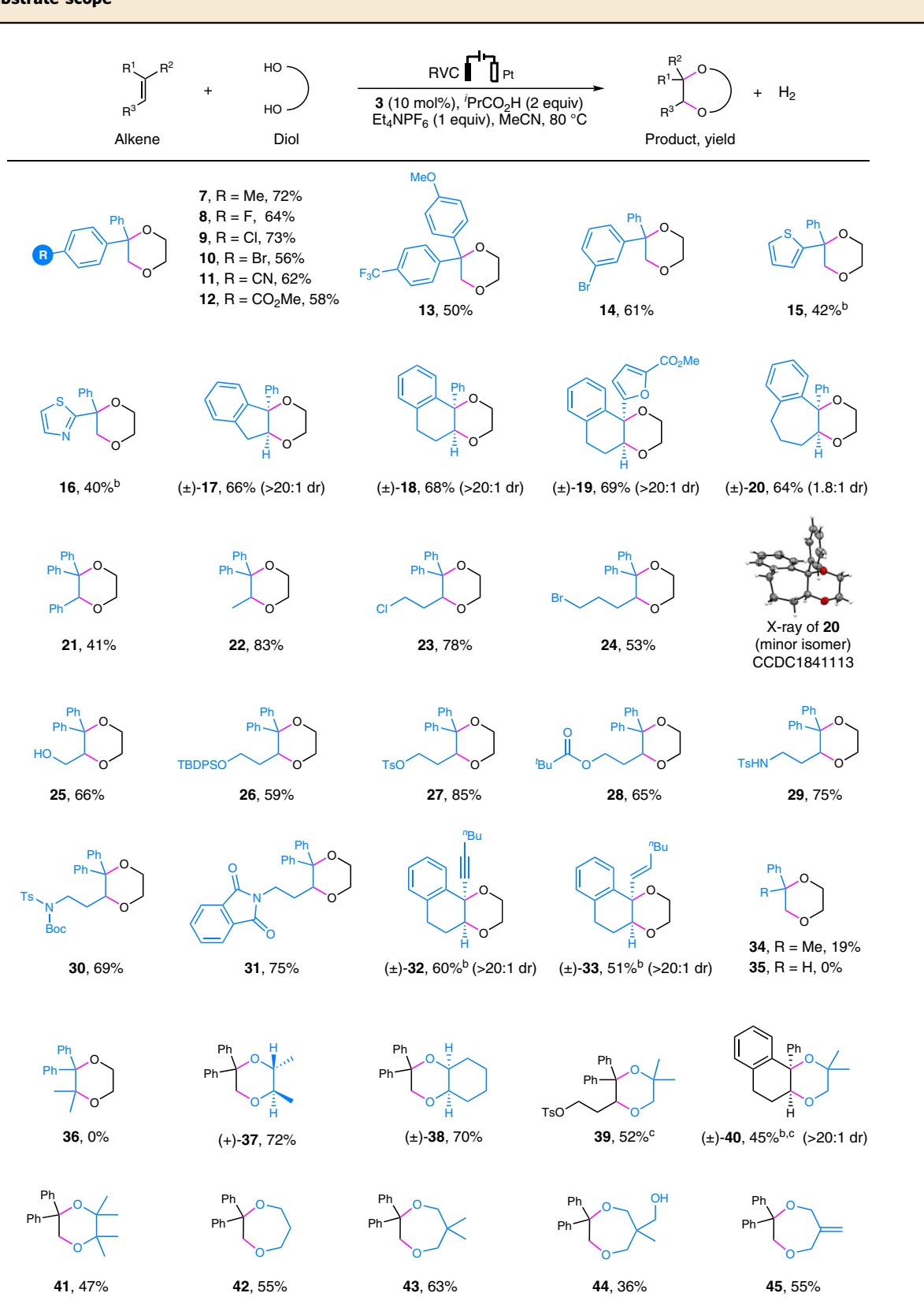

TBDPS, *tert*-butyldiphenylsilyl; Ts, *p*-toluenesulfonyl; Boc, *tert*-butyloxycarbonyl
[a]Reaction conditions: Alkene (0.2 mmol), diol (9 mmol), 1.7–4.5 h. All yields are isolated yields
[b]MeCN/CH₂Cl₂ (1:2) as solvent
[c]18 mmol of diol was employed

the corresponding radical cation **II** and regenerate **3**. The nucleophilic trapping of **II** with ethylene glycol (**2**) and the subsequent deprotonation produce the carbon-centered radical **III**[39,40,61–63], which is then oxidized by **I** to afford the carbon cation **IV**. The cyclization of **IV** eventually generates the 1,4-dioxane product **4** ($E_{p/2} = 1.95$ V vs SCE). On the cathode, protons are reduced to produce $H_2$. The addition of $^iPrCO_2H$ facilitates $H_2$ evolution and probably also plays an important role in reducing unwanted reduction of **I**, the $CH_2Cl_2$ solvent and the alkene substrate. The catalytic role of **3** was confirmed by the detection of a catalytic current[47,64] using cyclic voltammetry (Fig. 3b). The inclusion of **3** was also found to inhibit the oxidative decomposition of **4**, probably because of the inefficient oxidation of **4** by **3**-derived radical cation **I** (Supplementary

Fig. 1). This was supported by the observation that **4** remained largely stable when subjected to electrolysis in the presence of a catalytic amount of **3** (Fig. 4a). In contrast, under similar conditions but in the absence of **3**, 32% of **4** was found to have decomposed, resulting in the formation of 1,3-dioxane **5** and benzophenone **6** in 16 and 7% yields, respectively (Fig. 4b).

In summary, we have developed a triarylamine-catalyzed electrochemical annulation reaction for the synthesis of 1,4-dioxane and 1,4-dioxepane scaffolds from alkenes and diols. The reaction is compatible with a wide variety of functional groups and showed excellent tolerance for di- and trisubstituted alkenes, allowing facile access to functionalized O-heterocycles with tetrasubstituted carbon centers. We are currently investigating whether our alkene annulation reaction can be applied to the synthesis of other 1,4-heterocyclic compounds.

## Methods

**Representative procedure for the synthesis of 4**. A 10-mL three-necked round-bottomed flask was charged with **3** (0.02 mmol, 0.1 equiv), the alkene **1** (0.2 mmol, 1 equiv), ethylene glycol **2** (9 mmol, 45 equiv), $^iPrCO_2H$ (0.4 mmol, 2 equiv), and $Et_4NPF_6$ (0.2 mmol, 1 equiv). The flask was equipped with a reflux condenser, a RVC anode (100 PPI (pores per inch), ~ 65 $cm^2$ $cm^{-3}$, 1 cm × 1 cm × 1.2 cm) and a platinum plate (1 cm × 1 cm) cathode before being flushed with argon. Then, anhydrous MeCN was added. Constant current (12.5 mA, $j_{anode} = 0.16$ mA $cm^{-2}$) electrolysis was performed at reflux (internal temperature, 80 °C) until the alkene

**Fig. 2** Electrochemical gram scale reaction. Gram scale synthesis of **27**

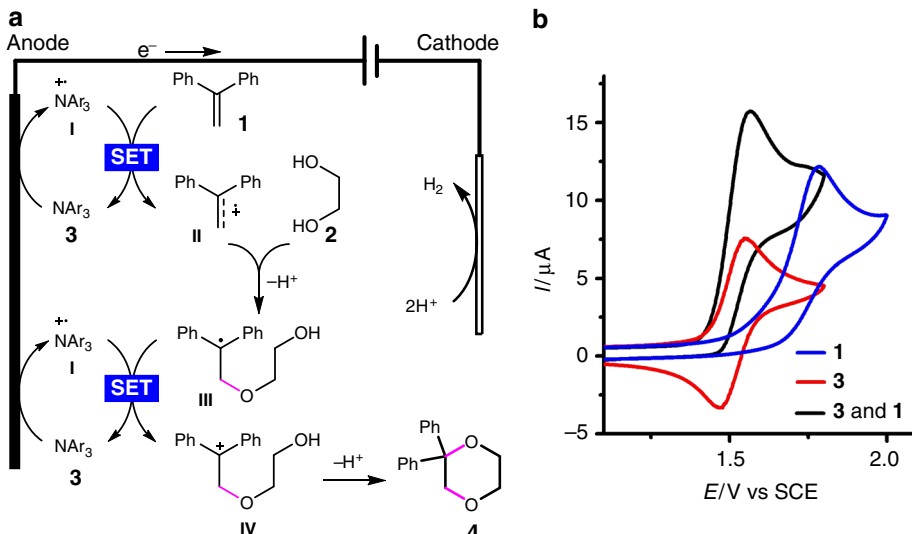

**Fig. 3** Mechanistic rationale and cyclic voltammograms. **a** Mechanistic proposal. **b** Cyclic voltammograms recorded in MeCN/$CH_2Cl_2$ (6:1) with 0.1 M $Et_4NPF_6$ as the supporting electrolyte. **3** (2.6 mM). **1** (1.3 mM). SET, single-electron transfer

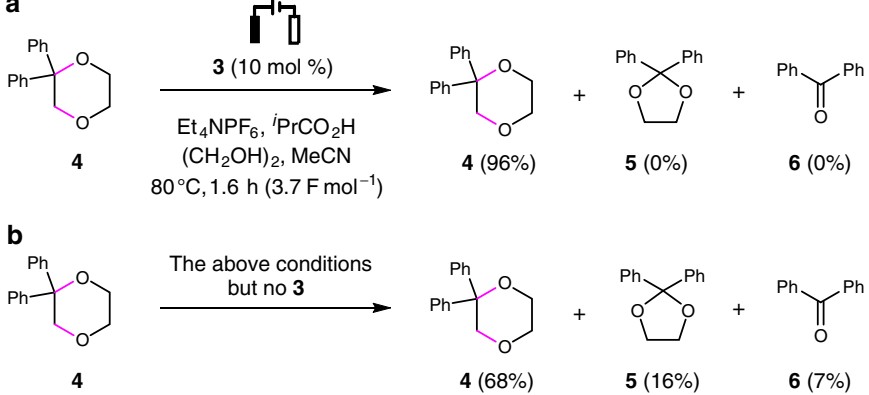

**Fig. 4** Electrolysis of compound **4**. **a** Electrolysis of **4** in the presence of triarylamine **3**. **b** electrolysis of **4** in the absence of **3**. Yields were determined by $^1H$ NMR using 1,3,5-trimethoxygenzene as the internal standard

substrate was completely consumed (monitored by thin layer chromatography or $^1$H NMR). The reaction mixture was cooled to room temperature (RT) and saturated $Na_2CO_3$ solution was added. The resulting mixture was extracted with EtOAc ($3 \times 20$ mL). The combined organic solution was dried with anhydrous $MgSO_4$ and concentrated under reduced pressure. The residue was chromatographed through silica gel and the product **4** was obtained in 91% yield as a white solid by eluting with ethyl acetate/hexanes. All new compounds were fully characterized (See the Supplementary Information).

**Procedure for the gram scale synthesis of 27**. The gram scale synthesis of compound **27** (80%, 2.5 g) was conducted in a 200-mL beaker-type cell with two RVC plates (100 PPI, 5 cm × 5 cm × 1.2 cm) as anode and a Pt plate cathode (3 cm × 3 cm) at constant current of 625 mA ($j_{anode} = 0.16$ mA cm$^{-2}$) for 0.9 h. The reaction mixture consisted of compound **2** (15 mL, 262 mmol), **46** (2.2 g, 5.8 mmol), **3** (0.42 g, 0.58 mmol), $^iPrCO_2H$ (1.1 mL, 11.6 mmol), MeCN (180 mL), and $Et_4NPF_6$ (1.6 g, 5.8 mmol).

## Data availability

The X-ray crystallographic coordinates for structures reported in this article have been deposited at the Cambridge Crystallographic Data Centre (CCDC), under deposition number CCDC 1841113 (**20**). The data can be obtained free of charge from The Cambridge Crystallographic Data Centre [http://www.ccdc.cam.ac.uk/data_request/cif]. The data supporting the findings of this study are available within the article and its Supplementary Information files. Any further relevant data are available from the authors on request.

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

## Acknowledgements

We acknowledge the financial support of this research from MOST (2016YFA0204100), NSFC (No. 21672178), the "Thousand Youth Talents Plan", and Fundamental Research Funds for the Central Universities.

## Author contributions

C.Y.C. performed the experiments and analyzed the data. H.C.X. designed and directed the project and wrote the manuscript.

## Additional information

**Competing interests:** The authors declare no competing interests.

