## [Peer Review File · Nature Communications]

Reviewer #1 (Remarks to the Author):

The paper by Xu et al reports an efficient electrochemical approach for the synthesis of functionalized 1,4-dioxanes and 1,4-dioxepanes via dehydrogenative annulation of alkenes with diols using triarylamine derivatives as the redox mediator. The broad scope range of substrates and excellent tolerance of functional groups are the main characteristic of the method. Especially, it is demonstrated that the mediator could inhibit the decomposition of the product, which is the first report. Therefore, I recommend its publication on Nature communication if the authors could address the following questions:

- 1) Table 1. It was found that only 5% yield of the desired product was generated using (4-BrC₆H₄)₃N as the catalyst, however, 58% yield was afforded in the absence of mediator. The results seem not logic, could the author explain the reasons?
- 2) Fig. 4. How to get the yield data? H NMR or isolated yield?
- 3) Line 63: the oxidation potential of (4-BrC₆H₄)₃N under the identical conditions is better to include.
- 4) Line 17: Additional?

Reviewer #2 (Remarks to the Author):

The manuscript by Xu's group reported a method for the synthesis of 1,4-dioxane and 1,4-dioxepane derivatives through an electrochemical dehydrogenative annulation reaction of alkenes with 1,2- and 1,3-diols. With the addition of the redox catalyst, triarylamine, the desired reactions proceeded smoothly (yield up to 91%). It's impressed that the gram scale synthesis was highly efficient, which means that this strategy has the potential for industrial application. However, the electrochemical dialkoxylation of olefins is not a new approach (see Ref 36, 37, 38, 38 and 40 in the manuscript) and in some cases, the dialkoxylation yields were reported as good as 80% above if the reaction time was carefully controlled. The major improvement of this work is the introduction of a redox catalyst, and thus the reaction occurred at a decreased potential and the diffusion problem was avoided on the electrode surface. Moreover, the substrate scope in this work was limited. Only 1,1-di-substituted olefins could yield the desired products, while the annulation of the styrene and tetra substituted alkenes failed to takeplace. Based on the comments above, I feel this work falls short on scholarly significance to be published in Nature Communications.

Following are two minor questions:

1. Given that the redox catalyst was involved in this reaction, is this protocol “reagent-free” as claimed in the title?
2. The note of d in Table 2 is missing.

Reviewer #3 (Remarks to the Author):

The manuscript by Xu and Cai presents a novel and innovative method for the transition metal-free annulation towards saturated heterocycles by electrocatalysis. Thus, the oxidant reagent-free annulation of alkenes was accomplished by different diols, delivering a variety of O-heterocycles. Thereby, the use of chemical oxidants was prevented, and thus also the formation of stoichiometric amounts of undesired byproducts could be avoided. While the substrate scope is somewhat limited to diarylalkenes and diols, the new method tolerates numerous important functional groups with ample scope. Mechanistic studies have been performed and a plausible mechanism has been proposed.

All new products were fully characterized, and the manuscript is accompanied by very good Supporting Information file.

Given the topical interest in efficient electrocatalysis, along with the sustainable nature of the metal-free annulation approach, I strongly recommend publication of this fine manuscript after minor revision.

- 1) The authors show the beneficial effect of the triarylamine as redox-mediator. The performance of other redox-mediators should be included in the main manuscript.
- 2) The improved chemo-selectivity of the reactions in the presence of the triarylamine redox-mediator are due to reduced degradation of the desired product 4, with $E_p/2 = 1.95$ V. Has a constant potential regime been tested to achieve the same goal in the absence of a redox-mediator?
- 3) What is the inherent diastereo-selectivity when using a typical chiral diol towards the alkenes (as for 7-12).
- 4) Table 2: Are nucleophilic primary amines or thiols tolerated?
- 5) Has the formation of H₂ been confirmed?
- 6) Recently published reviews on electrochemical C-H activation could be included.
- 7) Please, include the reaction temperature in the corresponding Schemes and Tables.

Point-by-point response to the referees' comments is as follows:

Reviewer #1

- The paper by Xu et al reports an efficient electrochemical approach for the synthesis of functionalized 1,4-dioxanes and 1,4-dioxepanes via dehydrogenative annulation of alkenes with diols using triarylamine derivatives as the redox mediator. The broad scope range of substrates and excellent tolerance of functional groups are the main characteristic of the method. Especially, it is demonstrated that the mediator could inhibit the decomposition of the product, which is the first report. Therefore, I recommendate its publication on Nature communication if the authors could address the following questions:

Response: We are grateful to the reviewer for the recommendation.

- 1) Table 1. It was found that only 5% yield of the desired product was generated using (4-BrC₆H₄)₃N as the catalyst, however, 58% yield was afforded in the absence of mediator. The results seem not logic, could the author explain the reasons?

Response: We thank the reviewer for raising this question. (4-BrC₆H₄)₃N is not an effective catalyst for the desired transformation because it has a relatively low oxidation potential. As a result, the conversion was low (65% unreacted starting material, information provided in Table 1) when (4-BrC₆H₄)₃N was used as a catalyst, leading to the low yield. Some of the starting alkene probably decomposed through non-oxidative pathways considering the inherent instability of styrenyl alkenes. On the other hand, the low yield using direct electrolysis was probably caused by decomposition of the product as indicted by the observation of byproducts derived from C-C bond cleavage.

- 2) Fig. 4. How to get the yield data? H NMR or isolated yield?

Response: The yields were determined using ¹H NMR using an internal standard. This information has been added to Fig. 4.

- 3) Line 63: the oxidation potential of (4-BrC₆H₄)₃N under the identical conditions is better to include.

Response: The oxidation potential has been included.

- 4) Line 17: Additional?

Response: We thank the reviewer for pointing out this grammatical error. We have changed it “In addition”.

Reviewer #2

- The manuscript by Xu's group reported a method for the synthesis of 1,4-dioxane and 1,4-dioxepane derivatives through an electrochemical dehydrogenative annulation reaction of alkenes with 1,2- and 1,3-diols. With the addition of the redox catalyst, triarylamine, the desired reactions proceeded smoothly (yield up to 91%). It's impressed that the gram scale synthesis was highly efficient, which means that this strategy has the potential for industrial application.

Response: We thank the reviewer for the positive comment. These reactions are rather easy to be scaled up. In our experience, the use of a redox catalyst is essential.

- However, the electrochemical dialkoxylation of olefins is not a new approach (see Ref 36, 37, 38, 38 and 40 in the manuscript) and in some cases, the dialkoxylation yields were reported as good as 80% above if the reaction time was carefully controlled. The major improvement of this work is the introduction of a redox catalyst, and thus the reaction occurred at a decreased potential and the diffusion problem was avoided on the electrode surface.

Response: Electrochemical alkene dialkoxylation are indeed known in the literature, although it is very limited in scope. These studies all employed direct electrolysis. Refs 36 and 37 (39 and 40 in the revised manuscript) studied intramolecular reactions of heteroatom-substituted alkenes. Refs 41-43 (previously refs 38-40) studied the electrochemical oxidation of aryl alkenes in methanol solution. While the cyclic alkenes (ref 43) afforded good yields of the dimethoxylated product, the acyclic alkenes (refs 41 and 42) usually give a mixture of products arising from methoxylation and dimerization even in pure methanol as the solvent and reactant. In ref 43, the reactions were usually terminated before full conversion to avoid oxidative decomposition of the dimethoxylated product. These limited investigations demonstrated that the electrochemical dialkoxylation is not trivial even using the alcohol reactant as the solvent.

The alkene radical cations are highly attractive intermediates for organic synthesis and electrochemistry is a powerful tool for their generation due to the tunability and reagent-free feature of electron transfer on the electrode. However, the radical cation generated on the electrode surface has a relatively high local concentration and is thus prone to dimerization, especially for those derived from terminal alkenes. In addition, the reactive intermediates generated on the electrode surface can react with the electrode leading to electrode passivation. The use of direct electrolysis often leads to overoxidation of the products as demonstrated in our manuscript and in ref. 43. As a result, the application of electrochemically generated radical cation for organic synthesis has been extremely limited.

Refs 39 and 40

Ref 43

Ref 42

Ref 41

In our work, we employed a redox catalyst that operates at a potential lower than the substrate and product, which protects the product from overoxidation. In addition, the use of a redox catalyst allows the formation of the radical cation in the bulk solution, which reduces dimerization and its reaction with the electrode. As pointed by Refree #1 and #3, we have developed an “efficient,

novel, and innovative method” for the preparation of saturated O-heterocycles, many of which remain to be difficult to access efficiently. Our strategy opens enormous opportunities for reaction discovery based on alkene radical cations, which are still under explored. Our concept for heterocycle synthesis has potentially broader implications for the growing field of organic electrochemistry.

- Moreover, the substrate scope in this work was limited. Only 1,1-di-substituted olefins could yield the desired products, while the annulation of the styrene and tetra substituted alkenes failed to takeplace. Based on the comments above, I feel this work falls short on scholarly significance to be published in Nature Communications.

Response: Our reactions tolerated 1,1-substituted and trisubstituted alkenes. The combination of trisubstituted olefins and substituted diols provide straightforward access to highly substituted 1,4-dioxane derivatives that are difficult for existing technologies. At present, the reactions work best with 1,1-diaryl substituted olefins as we have shown in this preliminary Communication. Aryl groups are essential subunits of bioactive compounds and materials. As show in Fig. 1 of the manuscript, several bioactive compounds contained a tetrasubstituted carbon center bearing two aryl groups. Our conceptionally novel approach allow, for the first time, 1,4-dioxane and 1,4-dioxepane derivatives to be prepared through straightforward annulation of alkenes and diols. Further investigations are ongoing in our laboratory to broaden the scope of the alkenes and to include other bisnucleophiles for the synthesis of other saturated heterocycles.

- Following are two minor questions:
 1. Given that the redox catalyst was involved in this reaction, is this protocol “reagent-free” as claimed in the title?

Response: We thank the reviewer for this question. We have changed “reagent-free” to “oxidizing reagent-free” to be more specific.

- 2. The note of d in Table 2 is missing.

Response: We thank the reviewer for point out this typo, which has been corrected. The note d should be c.

Reviewer #3

- The manuscript by Xu and Cai presents a novel and innovative method for the transition metal-free annulation towards saturated heterocycles by electrocatalysis. Thus, the oxidant reagent-free annulation of alkenes was accomplished by different diols, delivering a variety of O-heterocycles. Thereby, the use of chemical oxidants was prevented, and thus also the formation of stoichiometric amounts of undesired byproducts could be avoided. While the substrate scope is somewhat limited to diarylalkenes and diols, the new method tolerates numerous important functional groups with ample scope. Mechanistic studies have been performed and a plausible mechanism has been proposed.

All new products were fully characterized, and the manuscript is accompanied by very good Supporting Information file.

Given the topical interest in efficient electrocatalysis, along with the sustainable nature of the metal-free annulation approach, I strongly recommend publication of this fine manuscript after minor revision.

Response: We are grateful to the reviewer for the recommendation.

- 1) The authors show the beneficial effect of the triarylamine as redox-mediator. The performance of other redox-mediators should be included in the main manuscript.
- **Response:** We thank the reviewer for the suggestion. We have included two additional mediators in Table 1, one triarylamine derivative and one imidazole-derivative developed by Professor Little. These are the major two types of organic redox mediator reported in the literature. We have previously developed tetraarylhydrazine derivatives as the redox mediators. But those compounds have relative low oxidation potential and thus are not tested. Fortunately, the optimal mediator **3** is easily available and can be prepared in large scale.
- 2) The improved chemo-selectivity of the reactions in the presence of the triarylamine redox-mediator are due to reduced degradation of the desired product **4**, with $E_p/2 = 1.95$ V. Has a constant potential regime been tested to achieve the same goal in the absence of a redox-mediator?
- **Response:** We thank the reviewer for this suggestion. Product decomposition may be avoided under conditions with careful control of the electrode potential. But literature results show that some aryl alkenes give mainly dimerized products (ref 41) under constant potential electrolysis. Under constant

potential conditions, the initial current density is high, which would generate high concentration of radical cations, leading to dimerization products through radical cation-radical cation coupling or through radical cation-alkene coupling. In addition, constant potential experiments require a three-electrode configuration instead of the simple two-electrode setup for the constant current experiments and need more expensive instrument for controlling the potential. When constant potential is applied, the reactions take longer to complete and are more difficult to scale up. Moreover, generation of the reactive radical cation using direct electrolysis on the electrode surface often leads to electrode passivation (Little, R. D. *Chem. Soc. Rev.* **2014**, *43*, 2492; Stahl, S. S. *Chem. Sci.* **2018**, *9*, 356). Furthermore, a redox catalyst operates at potential lower than that of the starting material and can usually achieve better or different selectivity. Considering the above points, we did not test the reactions using constant potential.

- 3) What is the inherent diastereo-selectivity when using a typical chiral diol towards the alkenes (as for 7-12).

Response: We thank the reviewer for this question. We do not expect the reaction to be diastereoselective because the two aryl groups have basically the same steric properties, at least for the ones without ortho-substitution.

- 4) Table 2: Are nucleophilic primary amines or thiols tolerated?

Response: Unfortunately, we do not expect these groups to be tolerated because of their potential interference with the alkene radical cation and their relatively low oxidation potentials. We have tested one substrate bearing a tertiary amine, which decomposed under the reaction conditions and did not afford the desired product. However, amides or carbamates are expected to be tolerated.

- 5) Has the formation of H₂ been confirmed?

Response: Yes, the formation of H₂ has been confirmed by GC and by observation of gas bubbling on the cathode.

GC trace of H₂ from the gas in headspace of reactor.

- 6) Recently published reviews on electrochemical C-H activation could be included.

Response: We have added reviews on electrochemical C-H activation as refs 28-30

- 7) Please, include the reaction temperature in the corresponding Schemes and Tables.

Response: Reaction temperatures have been included in the schemes and tables.

Reviewer #1 (Remarks to the Author):

After carefully read the revised manuscript and the point by point response to all the comments, I further strengthen my initial judgement. Just what I mentioned before, in addition to the well-acceptable advantages using electrochemistry to replace conventional chemical oxidation, this paper provide an efficient means for the synthesis of 1,4-dioxane and 1,4-dioxepane derivatives, which are not easily accessible chemically or just obtained in lower yield under direct electrolysis. The broad substrate scope and tolerance of many common functional groups are also significant and promising for the development of potential bioactive lead compounds. In addition, the protocol could be easily scared up due to using undivided cell under constant current. It is worthy of noting, to the best of my knowledge, for the first time, the authors found that a mediator is able to inhibit the decomposition of the product. This concept may provide an alternative idea on how to improve the electrolytic efficiency of a particular reaction.

Based on aforementioned reasons, as well figuring out my questions, I strongly recommendate its publication in Nature commun.

Reviewer #3 (Remarks to the Author):

The revised manuscript has fully addressed all comments of the reviewers and can now be published as is.

Point-to-point response to reviewer comments

REVIEWERS' COMMENTS:

Reviewer #1 (Remarks to the Author):

After carefully read the revised manuscript and the point by point response to all the comments, I further strengthen my initial judgement. Just what I mentioned before, in addition to the well-acceptable advantages using electrochemistry to replace conventional chemical oxidation, this paper provide an efficient means for the synthesis of 1,4-dioxane and 1,4-dioxepane derivatives, which are not easily accessible chemically or just obtained in lower yield under direct electrolysis. The broad substrate scope and tolerance of many common functional groups are also significant and promising for the development of potential bioactive lead compounds. In addition, the protocol could be easily scaled up due to using undivided cell under constant current. It is worthy of noting, to the best of my knowledge, for the first time, the authors found that a mediator is able to inhibit the decomposition of the product. This concept may provide an alternative idea on how to improve the electrolytic efficiency of a particular reaction.

Based on aforementioned reasons, as well figuring out my questions, I strongly recommendate its publication in Nature commun.

Response: We are grateful to the reviewer for the nice comments of our work and for the recommendation.

Reviewer #3 (Remarks to the Author):

The revised manuscript has fully addressed all comments of the reviewers and can now be published as is.

Response: We are grateful to the reviewer for the recommendation.